# A Review of the Structure—Activity Relationship of Natural and Synthetic Antimetastatic Compounds

**DOI:** 10.3390/biom10010138

**Published:** 2020-01-14

**Authors:** Su Ki Liew, Sharan Malagobadan, Norhafiza M. Arshad, Noor Hasima Nagoor

**Affiliations:** 1Institute of Biological Science (Genetics and Molecular Biology), Faculty of Science, University of Malaya, 50603 Kuala Lumpur, Malaysia; 2Centre for Research in Biotechnology for Agriculture (CEBAR), University of Malaya, 50603 Kuala Lumpur, Malaysia

**Keywords:** substituent, synthesis, antimetastasis, structure-activity relationships, biotechnology

## Abstract

There are innumerable anticancer compounds derived from either natural or synthetic origins. Many of these compounds have been further developed through structural modifications to not only inhibit cancer cell growth but also to exert an antimetastatic effect. This is achieved by attaching different substituents to generate different structure—activity relationships. This review highlights the effectiveness of different functional groups known to have antimigration and antiproliferation activities, such as fluoro, methoxy, methyl, amino, hydroxy, nitro, bromo, chloro, methylamino, ethoxy, carbonyl, iodo, and trifluoromethyl groups. Additionally, the positioning of these functional groups plays an important role in their anticancer activities, which was evident in one of our studies comparing analogues of a natural compound. Thus, this review suggests future recommendations for the design and development of improved anticancer drugs with higher efficacy.

## 1. Introduction

Cancer, a chronic disease, ranks in the top three leading cause of death worldwide [1]. In the development of cancer treatment, natural and synthetic compounds have both been explored for their cytotoxicity [2]. Many Food and Drug Administration (FDA)-approved anticancer drugs, such as paclitaxel, vincristine, vinblastine, and bortezomib, are derivates of natural compounds sourced from various plants [3,4,5]. Consequently, more natural compounds have been discovered and reported for their anticancer activity, such as the epipodophyllotoxin derivatives, maytansine, bruceantin, thalicarpine, camptothecin, and lapachol [6]. Furthermore, modification of these compounds has proven to be more effective in minimising side effects and targeting other oncogenic phenotypes, such as metastasis. An important hallmark of cancer progression, metastasis is a complex cascade of events that involves separation of cancer cells from the primary tumour followed by intravasation, extravasation and the eventual formation of secondary tumours [7]. Due to the high mortality rate in cancer caused by metastasis [8], the development of antimetastatic drugs has become a new aim in modern cancer therapy.

While a number of literature reviews have addressed the structure–activity relationship (SAR) of anticancer agents in terms of inhibition of cancer cell growth, SAR on targeting the metastatic process has not received as much attention. Therefore, the aim of this review is to fill the gap in SAR analysis of different functional groups in natural product and synthetic analogues with their antimetastatic and antiproliferative properties. In this paper, the functional groups reviewed are the fluoro, methoxy, methyl, amino, hydroxy, nitro, bromo, chloro, methylamino, ethoxy, carbonyl, iodo and trifluoromethyl groups. They are classified as either effective or weak in antimigration and antiproliferation effects. Furthermore, the position of the functional groups may also affect the effectiveness of the substituent in blocking the migration and growth of cancer cells. 

## 2. Effective Functional Groups, Their Position and Antimigration Effects

### 2.1. Fluoro (F) Group

Abid et al. [9] investigated the antimetastatic activity of isocoumarin analogue using collagen type I invasion assay. Among the analogues, compound 3-(3’,4’-difluorophenyl) isocoumarin **1** was found to be the most potent in exerting antimetastatic activity compared to the control. SAR study (Figure 1) concluded the fluoro groups on the *meta* and *para* positions of the phenyl ring at C-3 and the double bond of the isocoumarin nucleus increases the antimetastatic effect. 

A series of brartemicin analogues were synthesised by Jiang et al. [10]. These analogues were the product of Mitsunobu coupling of the secondary hydroxyls benzyl protected α,α-D-trehalose with benzoic acid derivatives and functional group modification and deprotection. Anti-invasion activity of these synthetic analogues was assessed on colon cancer 26-L5 cells (Figure 2). Among these compounds, the 2,6-difluoro- substituted analogue **3h** maintained the anti-invasive activities (Table 1). However, the fluoro group at the 4-position of the benzoic acid ring coupled with the 3-methoxy group lost their anti-invasive ability, indicating that the activity of fluoro group is reduced during coupling with the functionally weak methoxy group. 

Focal adhesion kinase (FAK) is one of the most common intracellular kinases that regulate signalling pathways associated with cellular migration, proliferation, and survival [11], making it an important target in developing anticancer drugs. To produce better FAK inhibitors, Zhang and his team [12] designed and synthesised a series of new 1,3,4-oxadiazole derivatives possessing benzotriazole moiety (Figure 3). Among the analogues, fluoro-substituted compound **6** displayed the best FAK inhibitory activity and performed better than the reference drug, cisplatin (Table 2). Moreover, analysis of the positions of fluoro substituent in the compounds revealed that the *ortho*-substituted compound **6** has better inhibitory activity compared to the *meta*- substituted compound **8** and *para*-substituted compound **10**. 

Cathepsins, the cysteine proteases involved in the progression of various human cancers, have been shown to be promising therapeutic targets in cancer treatment [13]. For example, inhibition of one of its members, cathepsin L, reduced cancer cell invasion and migration [14]. Benzoylbenzophenone thiosemicarbazone analogues were synthesised and tested as potential cathepsin L inhibitors [15]. Among the derivatives (Figure 4), compound **12** (3-benzoylbenzophenone thiosemicarbazone) was able to inhibit the activity of cathepsin L significantly at a half maximal inhibitory concentration (IC_50_) value of 9.9 nM (Table 3). Besides, among the *para* substituted analogues, analogue **13** showed significantly high anti-cathepsin L activity similar to the unsubstituted analogue **12**. Moreover, 1,3-bis(2-fluorobenzoyl)-5-bromobenzene thiosemicarbazone **18** showed higher inhibition of cathepsin L with an IC_50_ value of 8.1 nM. Once again, the findings showed that the attachment of fluoro substituents improved antimigration activity. 

### 2.2. Methoxy (OCH_3_) Group

The epidermal growth factor receptor (EGFR) is one of the transmembrane receptor tyrosine kinase ErbB family [16]. It plays crucial roles in regulating cell proliferation [17], apoptosis [18], and migration [19]. As such, overexpression and/or mutation of EGFR are associated with the formation of malignant cells [20]. The activation of EGFR also stimulates vascular endothelial growth factor (VEGF), which helps to induce tumour angiogenesis [21]. The main steps in the induction of angiogenesis are mediated via a specific VEGF receptor, VEGFR-2 [22]. Therefore, EGFR and VEGFR-2 are important targets in cancer therapy, especially to inhibit metastasis and angiogenesis. In order to discover better anticancer agents, a series of 4-anilino-quinazoline derivatives were synthesised and tested for EGFR and VEGFR-2 inhibitory activities (Figure 5) [23]. The data (Table 4) showed that analogues with 6,7-dimethoxy substituent, such as **20**, **21b** and **21e**, have better inhibitory effects against EGFR and VEGFR-2 compared to the corresponding analogues **21a**, **21c**, **21f**, which were replaced by a dioxolane ring.

In another study, the compound (*E*)-6-methoxy-3-(4-methoxyphenyl)-2-[2-(5-nitrofuran-2-yl)vinyl]quinoline **22** showed weak cytotoxicity in all of the cancer and normal cell lines investigated but had the ability to inhibit cell migration and invasion [24]. Therefore, it can be deduced that the quinoline ring with C-6 methoxy group substitution contributes greatly to inhibit metastasis (Figure 6). Furthermore, combretastatin A4 (CA-4), a known antiangiogenesis agent, has also been found to contain methoxy groups [25]. Thus, methoxy groups are potential contributors to the antimetastatic effects.

Transendothelial migration and invasion of tumour cells through the vascular endothelial cell layer are crucial steps in metastasis formation [26]. Thus, Zhou and his team evaluated the potential of a series of 4-methyl-2-(4-pyridinyl)thiazole-5-carboxamide derivatives in inhibiting HUVEC cells migration (Figure 7) [27]. Based on their results (Table 5), analogue **23** exhibited approximately twice the IC_50_ value compared to **24a** (IC_50_ = 6.0 ± 1.6 vs. 3.4 ± 0.2 μM). This suggested the antimigration effect is higher in compounds with substitution of phenyl (R_7_) by electron-donating groups, such as methoxy, compared to those with electron-withdrawing groups, such as chloro and nitro.

In a study by Wu et al., EF24 analogues were analysed against lung cancer cell lines for their anticancer ability (Figure 8) [28]. Notably, compound **26** with three methoxy groups attached to its side showed greater cytotoxicity than EF24, **25** (Table 6). Furthermore, compound **26** also exhibited significant antimigration effect against A549 cells. Together, the findings from these studies indicate the importance of methoxy group in the antimetastatic activity of anticancer compounds.

### 2.3. Methyl (CH_3_) Groups

The methyl group is another functional group that has been identified to be essential for the anticancer effects of a compound (Figure 9). In a study by Miyanaga et al., when methyl groups are substituted at the dibenzodiazepinone core of BU-4664L, both anti-invasive and antiangiogenic activities were significantly increased (Table 7) [29]. Notably, the methylated compound **27** exhibited a significantly higher antimigration effect of human umbilical vein endothelial cells (HUVEC) with an IC_50_ value of 7.6 μg/mL (=15 nM). Moreover, this analogue also displayed a remarkable antiangiogenic effect with an IC_50_ value of 0.11 μg/mL. 

Isomalyngamides are the secondary metabolites isolated from the marine cyanobacterium *Lyngbya majuscule* [30]. Chang and his group synthesised the analogues of isomalyngamide A and further examined the effectiveness of the compounds against tumour cell migration [31]. The base-sensitive methylene proton (H6’) was replaced with a methyl group, a substitute important to restrict chemical alkylation, in the two analogues, **28** and **29** (Figure 10). Although both analogues did not affect MDA-MB-231 cell proliferation at 50 µM, they completely inhibited cell migration with IC_50_ values of 22.7 µM and 29.9 µM, respectively (Table 8). Hence, this finding highlighted that methyl group at H6’ is critical for the inhibition of cancer cell migration.

Dao and colleagues reported the antiangiogenic activity of novel diarylamino-1,3,5-triazine analogues on HUVEC cells (Figure 11) [32]. Since FAK is related to the antiangiogenic activity, the inhibition of FAK was evaluated upon treatment with the compounds. SAR analysis for various substitutions at the position R on the triazine ring in the compounds showed that attachment of a methyl group at compound **32** increased the inhibitory potency substantially, as compared with compound **30** (Table 9). 

### 2.4. Amino Group (NH_2_)

A study of 4-anilino-quinazoline derivatives for EGFR and VEGFR-2 inhibitory activities was performed by Barbosa et al. in 2013 (Figure 5) [23]. The presence of a hydrogen bond donor at the *para* position of the aniline moiety translated to compounds with significantly lower IC_50_ values, especially compounds **21d** (IC_50_ = 2.37 μM for EGFRwt and 1.02 μM for VEGFR-2) and **21g** (IC_50_ = 0.90 μM for EGFRwt and 1.17 μM for VEGFR-2) with the attachment of amino group (Table 4). The findings suggest that the amino group’s ability to donate hydrogen bond is crucial for EGFR and VEGFR-2 inhibitory activities.

Newly synthesised triarylethylene analogues were analysed for anticancer activity against breast cancer cell lines (Figure 12) [33]. Among the compounds, analogue **33** with an attached amino group exhibited significant enhancement of cytotoxicity with lower IC_50_ values compared to tamoxifen and ospemifene against MCF-7 and MDA-MB-231 breast cancer cell lines (Table 10). To further verify whether these analogues show anti-invasive and antimetastatic effects on MDA-MB-231, Kaur et al. analysed the in vitro antimigration activity and the expression levels of proteins related to adhesion, migration and metastasis [33]. They identified compound **33** attached with an amino group as the most effective analogue (Table 10). Hence, the amino substitution on triarylethylene analogues is vital for the enhancement of antiproliferation and antimetastatic activities.

Matrix metalloproteinases (MMPs) are related to cancer invasion and metastasis [34], and overexpression of MMPs causes breakdown of extracellular matrix (ECM), which can promote tumour invasion [35]. Members of this family, MMP-2 and MMP-9, are also critical in stimulating angiogenesis of tumour cells [36]; therefore, they are regarded as suitable targets for anticancer drugs [37]. Song et al. evaluated the inhibitory activities of synthetic benzamide Ilomastat analogues against MMP-2 and MMP-9 (Figure 13) [38]. Among these analogues, **35a** derivative substituted with amino group at the 2-position exhibited significant higher inhibition (IC_50_ = 0.19 nM) of MMP-2 compared to Ilomastat, **34** (IC_50_ = 0.94 nM) (Table 11). However, when the 2-amino group was modified to the 3-position or was acylated, both analogues lost potency against MMP-2 compared to **35a**, and yet, they showed better inhibition against MMP-9. Furthermore, modifying the substituent at the 4-position caused a lower inhibition of MMP-2, either through the introduction of an electron-donating or electron-withdrawing group. Thus, the inhibition of MMP-2 was not only influenced by the substitution of the amino group at the 2-position but also affected by the substituents at the 4-position. 

### 2.5. Hydroxy (OH) Group

SAR study on brartemicin analogues conducted by Jiang et al. [10] showed that anti-invasive activity was moderately active when hydroxy group was substituted at the 2- or 4-position of the benzoic acid ring. Both hydroxyl substituted analogues **3d** and **4** maintained the anti-invasive activity at an IC_50_ of not more than 1.0 µg/mL, although with slightly less potency compared to the natural compound **2** (Table 1).

### 2.6. Nitro (NO_2_) Group

In the examination of 4-methyl-2-(4-pyridinyl)thiazole-5-carboxamide derivatives in inhibiting HUVEC cells migration by Zhou and colleagues, it was found that analogue **24a** exhibited significantly stronger inhibition compared to analogue **23** (IC_50_ = 3.4 ± 0.2 vs. 6.0 ± 1.6 μM) (Table 5) [27]. Thus, the findings suggest that phenyl (R_7_) substitution with the electron-withdrawing nitro groups results in a weaker antimigration effect than electron-donating groups such as methoxy. 

### 2.7. Bromo (Br) Group

Wu et al. synthesised a series of 2,3-diaryl-4-thiazolidinone analogues and evaluated their effects against tumour cell proliferation and migration [39]. Interestingly, most of the 2-(3-(arylalkyl amino carbonyl) phenyl)-3-(2-methoxy-phenyl)-4-thiazolidinone derivatives showed high antiproliferation effect against non-small lung cancer cell line A549 and breast cell line MDA-MB-231. Among these analogues, compound **36** with the bromo substituent attached to the phenylethyl amino group (Figure 14) exhibited the highest antimigration effect with an IC_50_ of less than 0.05 mM (Table 12). These results strongly suggest that the bromo group is an important structural requirement to inhibit the migration of cancer cells.

### 2.8. Chloro (Cl) Group

The protein STAT3 is a member of the signal transducers and activators of transcription (STATs) family [40], that helps to regulate cell proliferation, apoptosis and metastasis [41]. Inhibition of STAT3 signalling has been demonstrated to prevent metastasis [42] and inhibit angiogenesis [43] in various tumour models, making it a therapeutic cancer target. To synthesise more potent STAT3 inhibitors, Gao and his team [44] developed a series of benzyloxyphenylmethylaminophenol analogues (Figure 15) and tested the inhibition of STAT3 signalling pathway using a STAT3 luciferase reporter method. Based on the experimental data (Table 13), compound **37b** with a chloro group attached at C-3 in ring B showed lower IC_50_ value (1.38 µM) compared to the compound without chloro group **37a** (IC_50_ = 7.71 µM). Moreover, changing the position of the chloro group from C-3 to C-5 reduced the inhibition the STAT3 activity. 

## 3. Weak Functional Groups, Their Position and Antimigration Effect

### 3.1. Methoxy (OCH_3_) Group

Jiang et al. [10] reported that the 2-methoxy substituted brartemicin analogue **3a** maintained its anti-invasive activity. The activity increased when methoxy group was substituted at both 2- and 3-positions, whereby the 2,3-dimethoxy-substituted analogue **2e** was more potent than 2,3-dihydroxyl **5** and the natural brartemicin (Table 1). However, the 4-methoxy-substituted **3c**, 3,4,5-trimethoxy-substituted **3f** and 3-methoxy-4-flurobenzoic esters **3g** completely lost the anti-invasive ability. Altogether, the results showed that the methoxy group is not a good substituent to the phenyl ring in inhibiting invasiveness.

Parker et al. in 2013 carried out the synthesis of benzoylbenzophenone thiosemicarbazone derivatives and assessed the inhibitory activity against cathepsins L [15]. The activity was diminished when the compound was substituted with a methoxy group. The authors suggested that the reduction in activity was caused by the increase in steric hindrance due to the substitution. As shown in Table 3, *p*-methoxy analogue **15** exhibited a high IC_50_ value of 5117 nM and is significantly weaker than unsubstituted analogue **12**. Thus, it was concluded that methoxy group does not have an important role in inhibiting cathepsins L.

Additionally, Limtrakul et al. reported the effects of curcumin **38**, demethoxycurcumin **39** and bisdemethoxycurcumin **40** (Figure 16) on the expressions of matrix metalloproteinases-2 (MMP-2), matrix metalloproteinases-9 (MMP-9), urokinase plasminogen activator (uPA), membrane Type 1 MMP (MT1-MMP), and tissue inhibitor of metalloproteinases (TIMP-2), in addition to in vitro invasiveness of human fibrosarcoma cells [45]. The compounds **39** and **40** had higher antimetastatic potency than **38** by differentially downregulating the extracellular matrix (ECM) degradation enzymes MMPs and uPA (Table 14). Based on the zymography results, **38**, **39** and **40** significantly decreased the cell secretion of uPA, active-MMP-2 and MMP-9 but not pro-MMP-2 in a dose-dependent manner. 

The MT1-MMP and TIMP-2 protein expressions reduced when treated with 10 μM with **39** and **40**, but the treatment with curcumin showed a slight reduction of MT1-MMP but not TIMP-2. Furthermore, these curcuminoids significantly inhibited activities of three enzymes, namely, collagenase, MMP-2 and MMP-9. The results concluded that the anti-invasion activity of the compounds can be ranked as **40** ≥ **39 > 38**. As shown in Figure 16, the antimetastatic potency of curcumin derivatives is increased by removal of one or two methoxy groups from the benzene ring. 

### 3.2. Methyl (CH_3_) Groups

Brartemicin analogues were studied for anti-invasion effect against murine colon 26-L5 carcinoma cells [10]. Among the various functional groups at the 2-position of the benzoic acid ring, 2-methyl substitution in analogue **3b** did not result in anti-invasive activity (Figure 2). Hence, methyl groups are not recommended as substituents for antimetastatic activity.

The FAK inhibitory effect of the 1,3,4-oxadiazole analogues was investigated [12]. The SAR analysis indicated that compounds with electron-donating groups showed weaker activity. An example is the methyl substitution in analogues **7**, **9** and **11**, with IC_50_ values in the range of 12.1–33.8 μM (Table 2). Their activity was significantly reduced compared to compounds with an electron-withdrawing group. Succinctly, methyl groups are unfavourable for FAK inhibition. 

Foudah et al. discussed the influence of different aromatic esters attached at the C-4 position of sipholenol A analogues on the antimigratory activity (Figure 17) [46]. They discovered that the presence of an electron-donating substituent, such as the methyl group at the *para*-position of the aromatic moiety (C-5’) of **41**, reduced the antimigratory activity when compared with an electron-withdrawing substituent, such as the fluoro group on **42** (Table 15).

### 3.3. Hydroxy (OH) Group

Benzoylbenzophenone thiosemicarbazone analogues were tested for the inhibition of cathepsins L [15]. It was observed that *para* substitution of hydroxy at the analogue **16** resulted in diminished anticancer activity. As shown in Table 3, the IC_50_ value of **16** was 340 nM and less potent than original analogue **12**. As such, the reduced activity was thought to be due to the steric hindrance by the *para* hydroxyl group. Hence, it is important to select a suitable substituent in order to reduce the steric hindrance effects on the anticancer activity.

Andrographolide derivatives were evaluated against cancer cell for antimigration and anti-invasion activities (Figure 18) [47]. Analogue **44**, in which the allylic hydroxyl at C-14 position was removed, had a better antimigration effect in human bladder carcinoma 5637 cells than the original compound (**43**) (Table 16). In other words, the hydroxy group may hinder the antimigration effect on cancer cells, which requires further validation.

### 3.4. Bromo (Br) Group

The benzoylbenzophenone thiosemicarbazone analogues **17** and **19** exhibited a remarkable decrease of cathepsin L inhibition among all the analogues. Both of these compounds have *m*-bromo substituents on the outermost rings of the benzoylbenzophenone molecular template [15]. Moreover, analogue **14** with *p*-bromo substituted bis-thiosemicabazone was unable to inhibit cathepsin L even at a high concentration of 10,000 nM (Table 3). Overall, the bromo substituent, which increases the steric bulk on the outermost rings, leads to the reduced inhibitory activity.

### 3.5. Chloro (Cl) Group

The antimigration activity of nine 4-methyl-2-(4-pyridinyl)thiazole-5-carboxamide analogues on HUVEC cells was examined [27], and the results showed that derivatives with substitution of phenyl (R_7_) by a chloro group have reduced antimigration effect when compared to the analogue substituted by a methoxy group. The data showed that antimigration activity of chlorinated analogue **24b** with IC_50_ value of 6.8 ± 2.3 was weaker than methoxylated analogue **24a** with IC_50_ value of 3.4 ± 0.2 (Table 5). 

### 3.6. Methylamino (NHCH_3_) Group

A study on the antiangiogenic activity of novel diarylamino-1,3,5-triazine analogues on HUVEC cells [32] showed that methylamino group attached at the position R on the triazinic ring of the compound **31** reduced the potency of FAK inhibition (Table 9). Hence, methylamino groups are not suitable as a substituent for antimigration activity. 

## 4. Effective Functional Groups, Their Position and Antiproliferation Effect

### 4.1. Chloro (Cl) Group

In 2001, Trapani et al. [48] synthesised and evaluated imidazobenzothiazole derivatives for cytotoxic activity against several cancer cells. A SAR study (Figure 19) concluded that the introduction of chloro group at the 7-position of the parent compound led to the enhancement of cytotoxic activity (Table 17). Furthermore, analogue **48** was derived from substitution with Cl atom at the 8-position of the monochloro derivative **45**, which displayed a minimum increase of activity. This finding suggested that the position of chloro substitution could affect antiproliferation activity. 

Morales and colleagues [49] examined the antiproliferation activity of purine derivatives against human breast, colon and melanoma cancer cell lines (Figure 20). Their results showed that 2,6-dichloropurine analogues **50** and **51** have the most potent activity against the assayed cell lines (Table 18). This SAR information draws attention to the importance of chloro substituents for the antiproliferation activity.

Several novel 8-hydroxyquinoline analogues were synthesised by Freitas et al., and the antiproliferation activity of the compounds were evaluated using several cancer cell lines (Figure 21) [50]. Aromatic rings which had halogen substituents such as fluorine, chlorine, bromine and iodine were chosen to study the electronic and steric impacts of the substituents. Among the halogenated derivatives, the chlorinated analogues **52c** and **52e** had the best antiproliferative effect (Table 19). The results proposed that chloro substituents exerted enhanced cytostatic activity. 

### 4.2. Methoxy (OCH_3_) Group

SAR analysis of (Z)-1-(1,3-diphenyl-1H-pyrazol-4-yl)-3-(phenylamino)prop-2-en-1-one derivatives revealed that a methoxy group as the electron-donating group attached to the aromatic B ring could contribute to significant cytotoxicity (Figure 22) [51]. Analogues **56**, **61**, **66**, **71** and **76**, with methoxy substitution at B ring, and analogues **57**, **62**, **67**, **72** and **77**, with 3,4,5,-methoxy substitution, showed better IC_50_ values than the analogues with an electron-withdrawing group (Table 20). 

Sreelatha et al. studied a series of novel naphthoquinone amide derivatives for anticancer activity against HeLa and SAS cancer cell lines [52]. Among the analogues synthesised (Figure 23), compounds with a methoxy substituent at C-2 of the quinone ring, such as **79a** and **79b**, were active (Table 21).

In contrast, the attachment of the methoxy group at the C-5 position reduced the antiproliferation activity of analogues **78a** and **78b**. Similar results were observed in our study on the anticancer activity of 1’*S*-1’-acetoxychavicol acetate (ACA) analogues on the MDA-MB-231 breast cancer cell line (Figure 24). The analogue 1’-acetoxy-3,5-dimethoxychavicol acetate with a methoxy group at the C-5 position showed significantly less activity compared to another analogue without the substituent, 1’-acetoxyeugenol acetate (Table 22) [53]. As such, not only is the methoxy group essential for anticancer activity, but the position of this group is also imperative in the effectiveness of the compound to inhibit the growth of cancer cells. 

### 4.3. Fluoro (F) Group

All of the synthesised 1,3,4-oxadiazole analogues were examined for anticancer activity against MCF-7 and HT29 cell lines [12]. It was noticeable that the fluorinated compound **6** showed the best activity against MCF-7 cells with an IC_50_ value of 5.68 µg/mL (Table 2), compared to the reference drug cisplatin with an IC_50_ value of 11.20 µg/mL. Besides, compound **6** also displayed better activity than the other analogues against HT29 cell with an IC_50_ value of 10.21 µg/mL. The SAR analysis illustrated that the fluoro group had the highest antiproliferation potency compared to other substituents. 

Zhou and coworkers designed, synthesised and evaluated the 6,7-disubstituted-4-phenoxyquinoline derivatives for in vitro cytotoxicity against various human cancer cell lines such as non-small cell lung cancer (A549), lung (H460), colorectal (HT-29), gastric (MKN-45), and glioblastoma (U87MG) (Figure 25) [54]. The introduction of an electron-withdrawing group, such as a fluoro group, to the analogue **81** led to an obvious improvement in anticancer activity (Table 23). Analogues **80**, **82** and **83** with at least one fluorine atom showed low IC_50_ values, indicating that the fluorine atom was necessary to improve the antiproliferation activity.

### 4.4. Methyl (CH_3_) Group

Newly synthesised naphthoquinone amide analogues were evaluated for their antiproliferation activity against HeLa and SAS cancer cell lines [52]. The introduction of methyl substituent at C-2 of the quinone ring of **79c** proved to be moderately effective for the antiproliferation activity (Table 21). 

In 2011, Zhang et al. evaluated the antiproliferation activity of a series of chalcone-type thiosemicarbazide analogues (Figure 26) [55]. Compound **84e** with a *para* methyl group in the B-ring exhibited the highest antiproliferation activity against HepG2 cells (Table 24). This demonstrated that the methyl group contributed to the potent anticancer activity.

### 4.5. Hydroxy (OH) Group

In 2016, (Z)-1-(1,3-diphenyl-1H-pyrazol-4-yl)-3-(phenylamino)prop-2-en-1-one derivatives were synthesised and analysed for anticancer effects against HT-29, PC-3, A549 and U87MG human cancer cell lines [51]. SAR analysis showed that hydroxy group, a strong electron-donating group, enhanced the activity of analogues **55**, **60**, **65**, **70**, and **75** (Table 20) when substituted on the aromatic B ring, with IC_50_ values of 1.35–3.21 µM. 

### 4.6. Ethoxy (CH_3_CH_2_O) Group

A series of novel (‒)-arctigenin analogues were synthesised and tested against human pancreatic cancer cell line PANC-1 for cytotoxicity (Figure 27) [56]. Among the (‒)-arctigenin analogues, monoethoxy analogue **86b** displayed the most preferential cytotoxicity (PC_50_) = 0.49 mM, followed by diethoxy analogue **86a** (PC_50_ = 0.66 mM), and triethoxy analogue **86c** (PC_50_ = 0.78 mM). In terms of potency, these compounds were either similar or more effective compared to (‒)-arctigenin (**85**) (PC_50_ = 0.80 mM) (Table 25). Thus, the introduction of the ethoxy group to the parent compound improved the cytotoxicity effect.

### 4.7. Carbonyl (C=O) Group

Gonçalves et al. tested the synthesised fluorinated asiatic acid analogues for antiproliferation activity against HeLa and HT-29 cell lines (Figure 28) [57]. Among the analogues, compounds **90**–**97** that had the pentameric A-ring with an α,β-unsaturated carbonyl showed significantly higher efficacy compared to the original asiatic acid **87** (Table 26). Also, the compounds exhibited lower IC_50_ values on HeLa cell line compared to the reference drug cisplatin. The findings are in accordance with previous results that proposed the introduction of α,β-unsaturated carbonyl moiety in A-ring of some triterpenes boosts the antiproliferation activity [58]. Furthermore, the significance of α,β-unsaturated carbonyl was subsequently proven by the reduced antiproliferation effect upon the conversion of analogue **90** into the nitrile analogue **98**.

## 5. Weak Functional Groups, Their Position and Antiproliferation Effect

### 5.1. Methoxy (OCH_3_) Group

To obtain data on the SAR in the imidazobenzothiazole series, Trapani et al. [48] examined several analogues for their cytotoxic effects. Introduction of electron-donating substituents such as methoxy group at C-7 position gave better activity compared to their parent compound. However, additional substitution of methoxy groups at the 5- and 8- positions (analogues **47** and **49**) caused the reduction of cytotoxic activity compared to the mono-methoxy analogue **46** (Table 17). Hence, the more methoxy groups are substituted to the compound, the less efficient is its ability to inhibit the growth of cancer cells.

Structure–activity relationships in the thiosemicarbazide derivatives synthesised by Zhang et al. [55] demonstrated that the analogues with changes to the methoxy group at *ortho* (**84a**), *meta* (**84b**) and *para* (**84c**) positions in the A-ring led to a remarkably reduced cytotoxic effect (Table 24). 

### 5.2. Bromo (Br) Group

Freitas et al. carried out a study and presented the effects of 8-hydroxyquinoline analogues on antiproliferation activity [50]. The brominated analogue **52d** showed weak cytotoxic effects with the log of molar concentration that inhibits 50% net cell growth (GI_50_) mean graph midpoint (MG MID) value = 1.3 (Table 19). In short, the bromo group attached to the compounds reduced the potency of antiproliferation.

In the Zhang et al. study, thiosemicarbazide analogues with halogen substitution at the *para* position in the B-ring mostly showed good antiproliferation activity [55]. However, analogue **84d** with a bromo substituent at the *para* position of the B-ring exhibited low activity (Table 24). 

### 5.3. Methyl (CH_3_) Group

Substitution of a methyl group at the benzene ring of 1,3,4-oxadiazole derivatives gave rise to low antiproliferation activity [12]. It is depicted clearly in Table 2, as the methylated analogues **7**, **9** and **11** had higher IC_50_ values than other analogues, which is in the range of 18.89–42.30 μg/mL. Hence, the methyl group is an unfavourable substituent in terms of anticancer activity.

### 5.4. Fluoro (F) Group

Compound **52b** in a series of 8-hydroxyquinoline derivatives [50] did not exhibit significant cytotoxicity due to the substitution of fluorine atom at the 4-position (Table 19). Hence, fluoro groups may not be a good choice for attachment to compounds for inhibiting cancer cell growth.

### 5.5. Iodo (I) Group

The analogue of 8-hydroxyquinoline **52a** with an iodine atom at the 4-position showed a low cytotoxic effect when compared to other halogen substituents [50] (Table 19). Iodine is considered a weak electronegative atom and might not increase the lipophilicity and hence will lower the antiproliferation activity of the compound.

### 5.6. Chloro (Cl) Group

Reddy and colleagues studied the antiproliferation effects of (*Z*)-1-(1,3-diphenyl-1H-pyrazol-4-yl)-3-(phenylamino)prop-2-en-1-one derivatives against various cancer cell lines [51]. The data showed that analogues **53**, **58**, **63**, **68** and **73** with an electron-withdrawing chloro substitution on the B ring exhibited low cytotoxic activity on the cancer cells (Table 20).

### 5.7. Trifluoromethyl (CF_3_) Group

From the SAR study of (*Z*)-1-(1,3-diphenyl-1H-pyrazol-4-yl)-3-(phenylamino)prop-2-en-1-one derivatives against different cancer cells [51], analogues **54**, **59**, **64**, **69**, and **74** with a trifluoromethyl group substituted on the B ring showed weak cytotoxic effects (Table 21). This indicated that the trifluoromethyl group is not important for antiproliferation activity.

### 5.8. Hydroxy (OH) Group

Evaluation of the SAR of novel fluorinated asiatic acid analogues was carried out based on their antiproliferation effect against HeLa and HT-29 cell lines [57]. As shown in Table 26, compound **88** with three free hydroxyl groups in the A-ring exhibited lower antiproliferation activity when compared with compound **89** which had two free hydroxyl groups. However, when the two hydroxyl groups in compound **89** were acetylated, the activity increased. These results indicated that free hydroxyl groups in A-ring are not important for the antiproliferation effect.

## 6. Conclusions

Discovery and development of anticancer agents that can inhibit metastasis is an important agenda in cancer therapy. As such, structural modifications of potential anticancer compounds can elucidate the functions of substitutions in mediating antimigration and antiproliferation effects. SAR analysis developed by different research teams is summarised in Table 27. Among the functional groups, most of the electron-withdrawing groups such as fluoro, chloro, nitro, amino, and carbonyl groups showed stronger activity than those with electron-donating groups such as methyl and methoxy groups. Although some studies showed that electron-donating groups performed well in anticancer activity while electron-withdrawing groups exhibited weak activity, this could be due to the influence of steric hindrance. Moreover, in terms of position of substituents, *para* substitution displayed better inhibition of both migration and growth of cancer cells. Overall, there is still a lot to explore in terms of the effects of different functional groups and their positions toward anticancer activity. With this in consideration, further investigation can be carried out to develop better anticancer drugs with improved antimetastatic activity. 

## Figures and Tables

**Figure 1 biomolecules-10-00138-f001:**
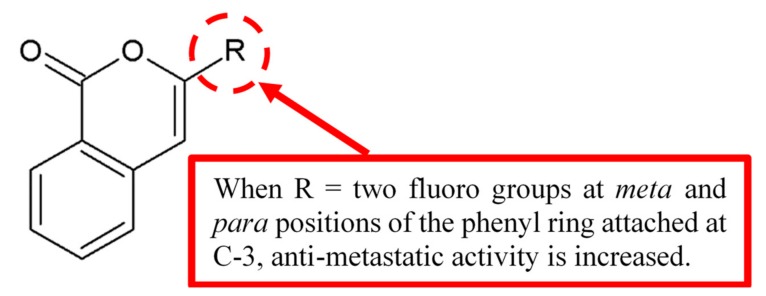
Structure–activity relationship (SAR) study of isocoumarin derivative **1**.

**Figure 2 biomolecules-10-00138-f002:**
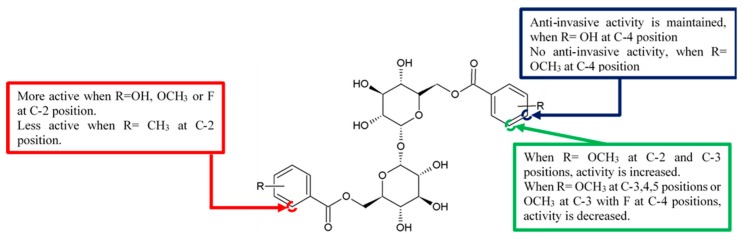
SAR study of brartemicin derivatives.

**Figure 3 biomolecules-10-00138-f003:**
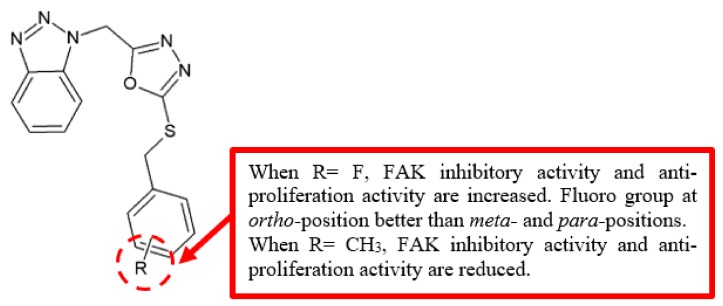
SAR study of 1,3,4-oxadiazole derivatives.

**Figure 4 biomolecules-10-00138-f004:**
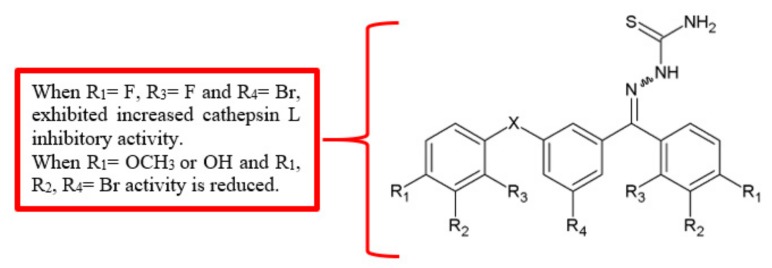
SAR study of benzoylbenzophenone thiosemicarbazone derivatives.

**Figure 5 biomolecules-10-00138-f005:**
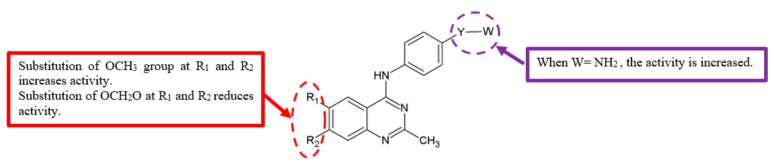
SAR study of 4-anilino-quinazoline derivatives.

**Figure 6 biomolecules-10-00138-f006:**
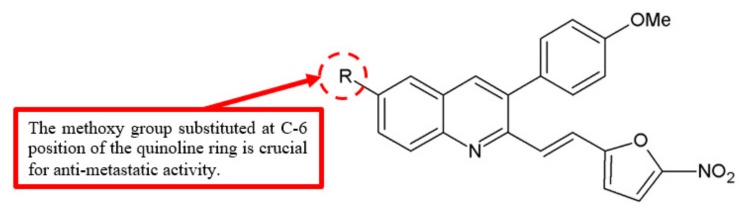
SAR study of 2-furanylvinylquinoline derivative **22**.

**Figure 7 biomolecules-10-00138-f007:**
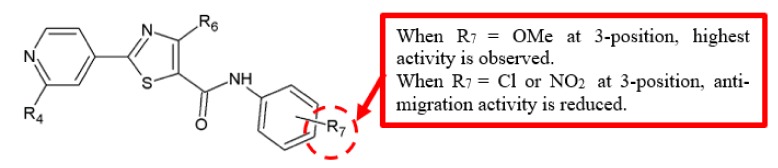
SAR study of 4-methyl-2-(4-pyridinyl)thiazole-5-carboxamide derivatives.

**Figure 8 biomolecules-10-00138-f008:**
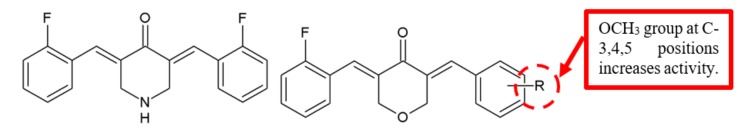
SAR study of EF24 derivatives.

**Figure 9 biomolecules-10-00138-f009:**
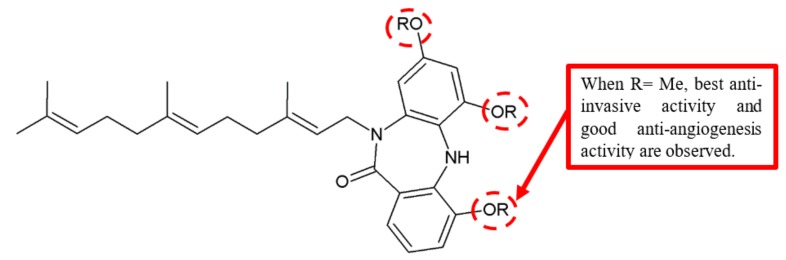
SAR study of BU-4664L derivative **27**.

**Figure 10 biomolecules-10-00138-f010:**
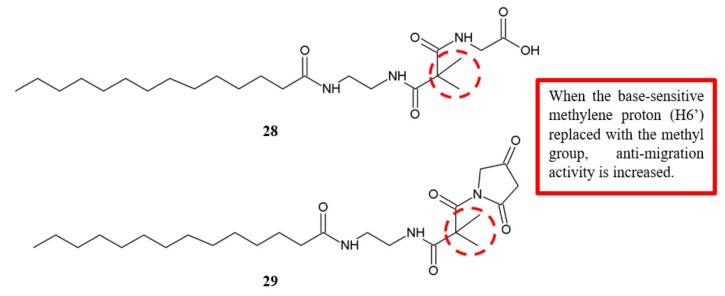
SAR study of isomalyngamide A derivatives.

**Figure 11 biomolecules-10-00138-f011:**
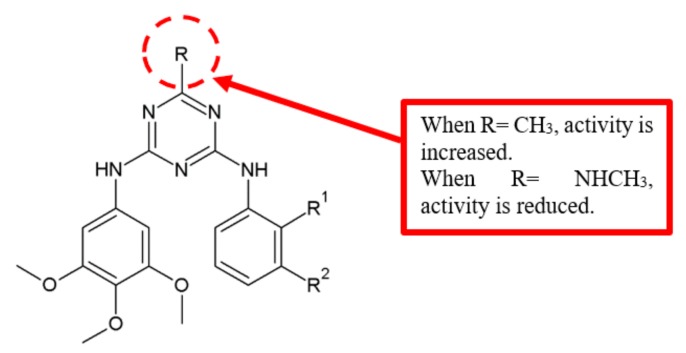
SAR study of diarylamino-1,3,5-triazine derivatives.

**Figure 12 biomolecules-10-00138-f012:**
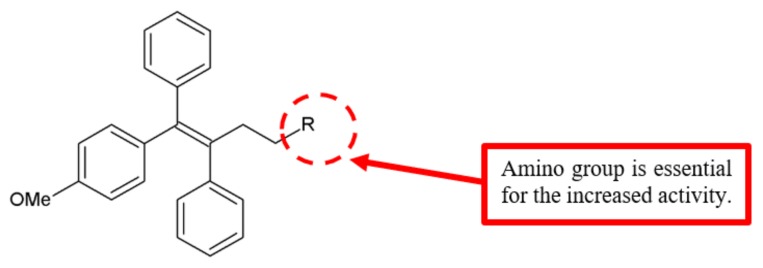
SAR study of triarylethylene derivatives.

**Figure 13 biomolecules-10-00138-f013:**
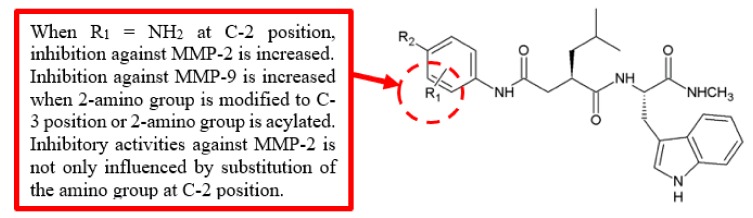
SAR study of benzamide Ilomastat derivatives.

**Figure 14 biomolecules-10-00138-f014:**
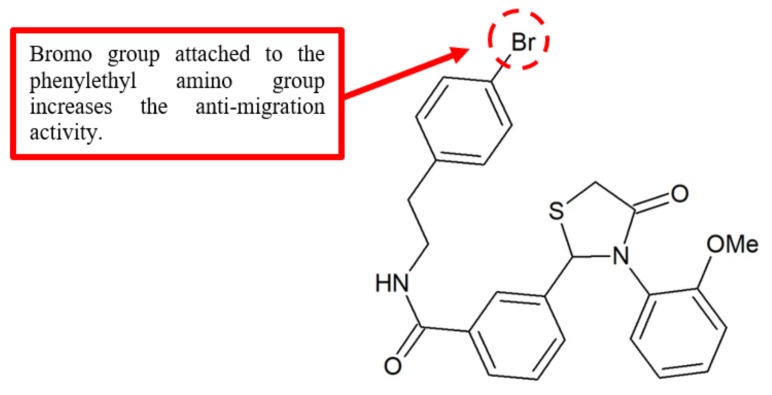
SAR study of 2,3-diaryl-4-thiazolidinone derivative **36**.

**Figure 15 biomolecules-10-00138-f015:**
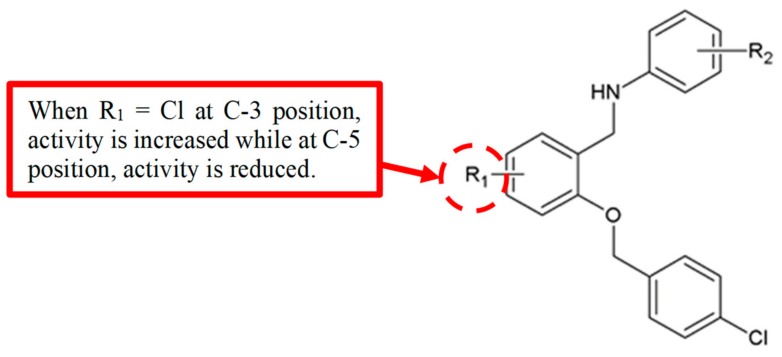
SAR study of benzyloxyphenylmethylaminophenol derivatives.

**Figure 16 biomolecules-10-00138-f016:**
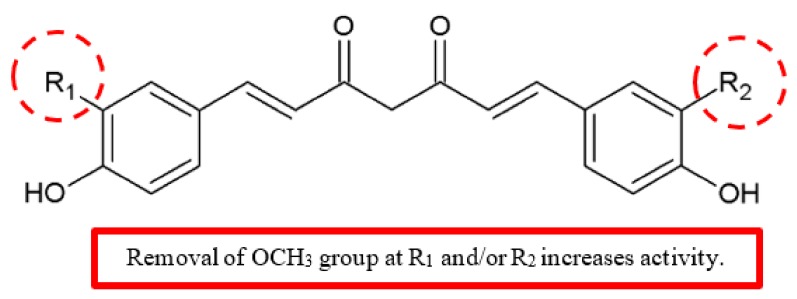
SAR study of curcumin derivatives.

**Figure 17 biomolecules-10-00138-f017:**
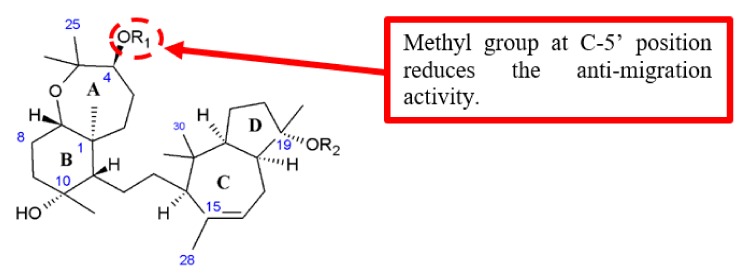
SAR study of sipholenol A derivatives.

**Figure 18 biomolecules-10-00138-f018:**
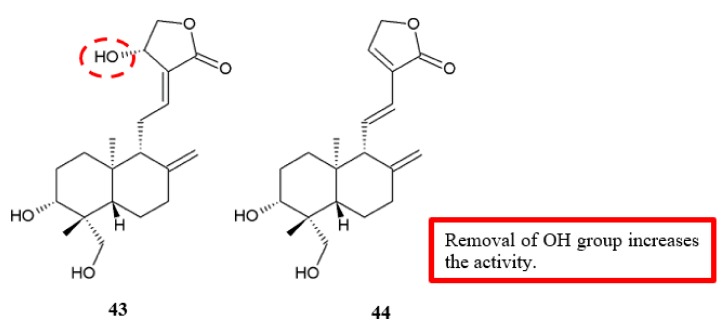
SAR study of andrographolide derivatives.

**Figure 19 biomolecules-10-00138-f019:**
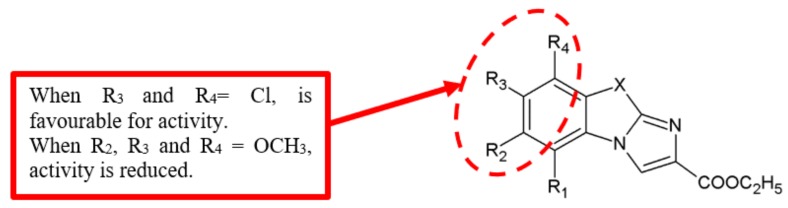
SAR study of imidazobenzothiazole derivatives.

**Figure 20 biomolecules-10-00138-f020:**
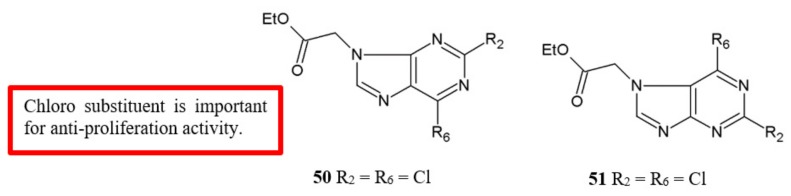
SAR study of purine derivatives.

**Figure 21 biomolecules-10-00138-f021:**
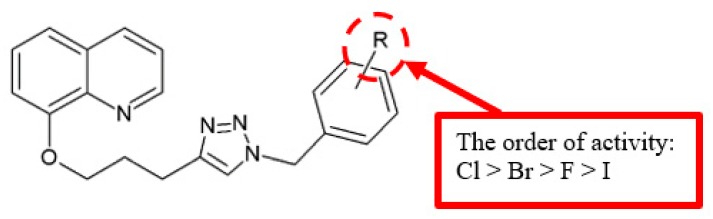
SAR study of 8-hydroxyquinoline derivatives.

**Figure 22 biomolecules-10-00138-f022:**
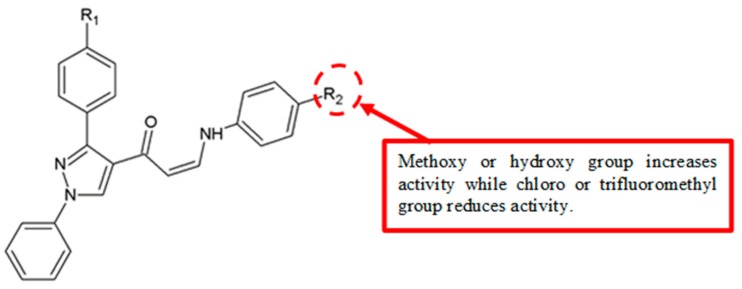
SAR study of (Z)-1-(1,3-diphenyl-1H-pyrazol-4-yl)-3-(phenylamino)prop-2-en-1-one derivatives.

**Figure 23 biomolecules-10-00138-f023:**
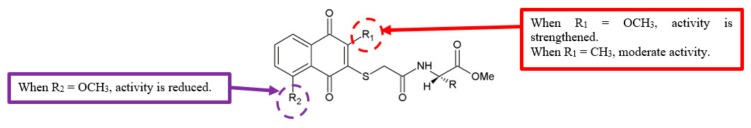
SAR study of naphthoquinone amide derivatives.

**Figure 24 biomolecules-10-00138-f024:**
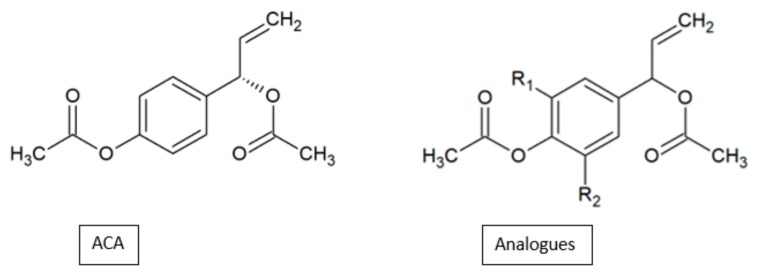
SAR study of 1’S-1’-acetoxychavicol acetate (ACA) derivatives.

**Figure 25 biomolecules-10-00138-f025:**
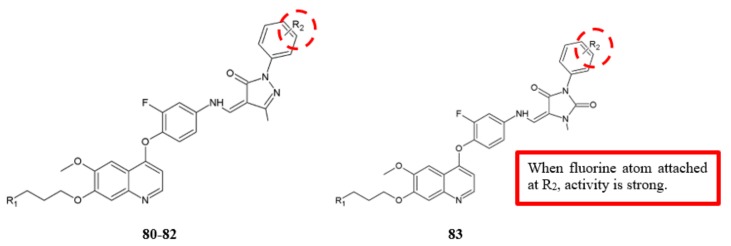
SAR study of 6,7-disubstituted-4-phenoxyquinoline derivatives.

**Figure 26 biomolecules-10-00138-f026:**
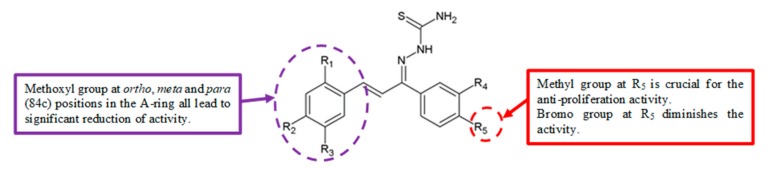
SAR study of chalcone thiosemicarbazide derivatives.

**Figure 27 biomolecules-10-00138-f027:**
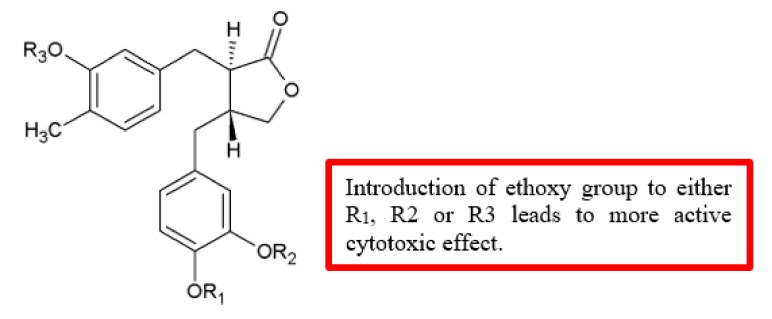
SAR study of (‒)-arctigenin derivatives.

**Figure 28 biomolecules-10-00138-f028:**
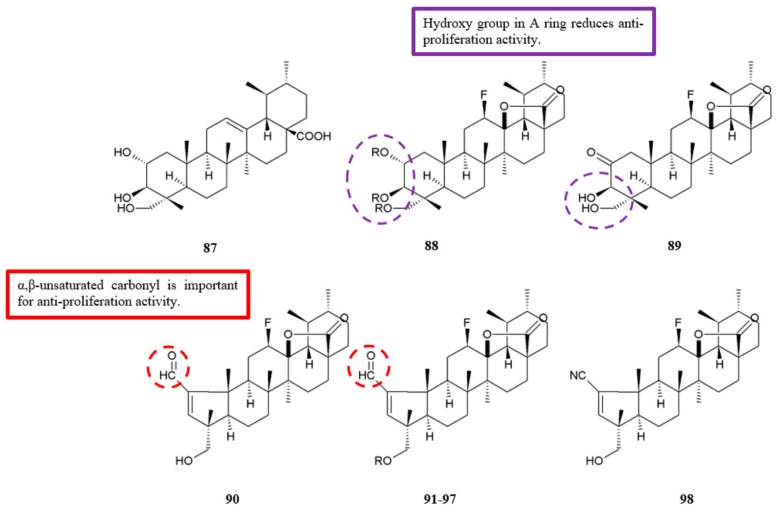
SAR study of asiatic acid derivatives.

**Table 1 biomolecules-10-00138-t001:** SAR study of brartemicin derivatives.

Compounds	R	IC_50_ (μg/mL)
		Anti-Invasive Activity In Vitro
Brartemicin, 2	2,4-(OH)_2_-6-CH_3_	0.25
**3a**	2-OCH_3_	1.0
**3b**	2-CH_3_	NA
**3c**	4-OCH_3_	NA
**3d**	4-OH	1.0
**3e**	2,3-(OCH_3_)_2_	0.10
**3f**	3,4,5-(OCH_3_)_3_	NA
**3g**	3-OCH_3_-4-F	NA
**3h**	2,6-F_2_	1.0
**4**	2-OH	<1.0
**5**	2,3-(OH)_2_	<1.0

**Table 2 biomolecules-10-00138-t002:** SAR study of 1,3,4-oxadiazole derivatives.

Compounds	R	Anti-Proliferation Activity (IC_50_, μg/mL)		FAK Inhibitory Activity (IC_50_, μM)
		MCF-7	HT29	
**6**	2-F	5.68	10.21	1.2 ± 0.3
**7**	2-CH_3_	18.89	26.81	12.1 ± 1.3
**8**	3-F	8.25	15.47	7.1 ± 0.3
**9**	3-CH_3_	28.92	38.50	15.8 ± 1.1
**10**	4-F	8.70	17.62	9.1 ± 0.5
**11**	4-CH_3_	30.23	42.30	33.8 ± 1.4
Cisplatin	-	11.20	15.83	8.6 ± 0.2

**Table 3 biomolecules-10-00138-t003:** SAR study of benzoylbenzophenone thiosemicarbazone derivatives.

Compounds	R_1_	R_2_	R_3_	R_4_	X	Cathepsin L Inhibitory Activity (IC_50_, nM)
**12**	H	H	H	H	C=O	9.9
**13**	F	H	H	H	C=O	14.4
**14**	Br	H	H	H	C=NHHC(S)NH_2_	>10,000
**15**	OCH_3_	H	H	H	C=O	5117
**16**	OH	H	H	H	C=O	340
**17**	H	Br	H	OH	C=O	~10,000
**18**	H	H	F	Br	C=O	8.1
**19**	H	Br	H	Br	C=O	10,347

**Table 4 biomolecules-10-00138-t004:** SAR study of 4-anilino-quinazoline derivatives.

Compounds	R_1_, R_2_	Y	W	Inhibitory Activity on EGFR (IC_50_, μM)	Inhibitory Activity on VEGFR-2 (IC_50_, μM)
**20**	OCH_3_, OCH_3_	SO_2_	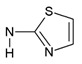	9.70	7.79
**21a**	OCH_2_O	SO_2_	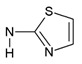	>100	>100
**21b**	OCH_3_, OCH_3_	SO_2_	CH_3_	61.5	>100
**21c**	OCH_2_O	SO_2_	CH_3_	>100	>100
**21d**	OCH_3_, OCH_3_	SO_2_	NH_2_	2.37	1.02
**21e**	OCH_3_, OCH_3_	-	N(CH_3_)_2_	36.0	39.3
**21f**	OCH_2_O	-	N(CH_3_)_2_	>100	>100
**21g**	OCH_3_, OCH_3_	C=O	NH_2_	0.90	1.17

**Table 5 biomolecules-10-00138-t005:** SAR study of 4-methyl-2-(4-pyridinyl)thiazole-5-carboxamide derivatives.

Compounds	R_4_	R_6_	R_7_	Anti-Migration Activity (IC_50_, μM)
				HUVEC
**23**	-	-	-	6.0 ± 1.6
**24a**	n-Pr	Me	3’-OMe	3.4 ± 0.2
**24b**	n-Pr	Me	3’-Cl	6.8 ± 2.3
**24c**	n-Pr	Me	3’-NO_2_	8.5 ± 6.7

**Table 6 biomolecules-10-00138-t006:** SAR study of EF24 derivatives.

Compounds	R	Anti-Proliferation Activity (IC_50_, μM)			Migration Rate at the Concentration of 20 µM (%)
		A549	LLC	H1650	A549
**EF24, 25**	-	7.1 ± 3.2	8.4 ± 3.0	14.6 ± 10	-
**26**	3,4,5-OCH_3_	6.3 ± 0.3	6.1 ± 0.9	6.8 ± 0.3	37.9

**Table 7 biomolecules-10-00138-t007:** SAR study of BU-4664L derivative **27**.

Compounds	Side Chain	R	IC_50_ (μg/mL)				
			Anti-Invasive Activity		Anti-Angiogenic Activity	Inhibition of Cellular Motility	
			Colon 26-L5	Renca		Colon 26-L5	HUVEC
**27**	Saturated	Me	1.0	0.78	0.11	0.67	0.0076

**Table 8 biomolecules-10-00138-t008:** SAR study of isomalyngamide A derivatives.

Compounds	Percentage of Cell Proliferation Inhibition at 20 μM (%)		Anti-Migration Activity (IC_50_, μM)
	MCF-7	MDA-MB-231	MDA-MB-231
**28**	7%	0%	22.7 ± 1.3
**29**	19%	0%	29.9 ± 0.6

**Table 9 biomolecules-10-00138-t009:** SAR study of diarylamino-1,3,5-triazine derivatives.

Compounds	R^1^	R^2^	R	FAK Inhibitory Activity—FRET (IC_50_, μM)	Anti-Proliferation Activity (IC_50_, μM)
				HUVEC	
**30**	NHSO_2_CH_3_	H	Cl	41.9 ± 4.6	9.5 ± 1.0
**31**	NHSO_2_CH_3_	H	NHCH_3_	65.9 ± 9.6	34.2 ± 7.6
**32**	NHSO_2_CH_3_	H	CH_3_	7.9 ± 0.9	8.5 ± 0.4

**Table 10 biomolecules-10-00138-t010:** SAR study of triarylethylene derivatives.

Compounds	R	Anti-Proliferation Activity (IC_50_, μM)		Migration Rate at the Concentration of 1 µM (%)
		MDA-MB-231	MCF-7	MDA-MB-231
**33**	NH_2_	11.4 ± 4.2	16.9 ± 7.7	25
**Tamoxifen**	-	>50	50	-
**Ospemifene**	-	>50	>50	>70%

**Table 11 biomolecules-10-00138-t011:** SAR study of benzamide Ilomastat derivatives.

Compounds	R_1_	R_2_	Inhibitory Activity (IC_50_, nM)	
			MMP-2	MMP-9
**Ilomastat, 34**	-	-	0.94	0.55
**35a**	2-NH_2_	H	0.19	1579.01
**35b**	2-NH_2_	F	2.20	7.75
**35c**	2-NH_2_	CF_3_	>10^4^	>10^4^
**35d**	2-NH_2_	COPh	>10^4^	>10^4^
**35e**	2-NH_2_	CH_3_	>10^4^	7297.04
**35f**	2-NH_2_	Br	21.80	27.32
**35g**	2-NHCOCH_3_	H	>10^4^	155.19
**35h**	3-NH_2_	H	2.05	13.52

**Table 12 biomolecules-10-00138-t012:** SAR study of 2,3-diaryl-4-thiazolidinone derivative **39**.

Compounds	Anti-Proliferation Activity (IC_50_, μM)		Anti-Migration Activity (IC_50_, μM)
	A549	MDA-MB-231	MDA-MB-231
**36**	0.21	0.23	<0.05

**Table 13 biomolecules-10-00138-t013:** SAR study of benzyloxyphenylmethylaminophenol derivatives.

Compounds	R_1_	R_2_	Inhibitory Activity on STAT3 (IC_50_, μM)	Anti-Proliferation Activity (IC_50_, μM)
			HepG2	MDA-MB-468
**37a**	H	4′-OH	7.71	9.61
**37b**	3-Cl	4′-OH	1.38	19.70
**37c**	5-Cl	4′-OH	26.68	18.83
**37d**	5-Cl	4′-SO_2_NH_2_	35.67	24.34

**Table 14 biomolecules-10-00138-t014:** SAR study of curcumin derivatives.

Compounds	R_1_, R_2_	IC_50_ (μM)				
		Inhibition of Active-MMP-2 Secretion	Inhibition of Active-MMP-9 Secretion	Inhibition of uPA Secretion	Inhibition of Collagenase Activity	Inhibition of MMP-2 Activity
**Curcumin, 38**	OCH_3_, OCH_3_	9.0	>10.0	10.0	50.0	>50.0
**Demethoxycurcumin, 39**	OCH_3_, -	6.0	8.0	7.5	47.0	45.0
**Bisdemethoxycurcumin, 40**	-, -	7.0	>10.0	7.0	>50.0	40.0

**Table 15 biomolecules-10-00138-t015:** SAR study of sipholenol A derivatives.

Compounds	R_1_	R_2_	Anti-Proliferation Activity (IC_50_, μM)	Anti-Migration Activity (IC_50_, μM)
MCF-7	MDA-MB-231	MDA-MB-231
**41**	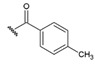	H	>50	33.4	11.8
**42**	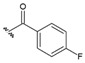	H	33.5	11.3	2.4

**Table 16 biomolecules-10-00138-t016:** SAR study of andrographolide derivatives. Im (%) refers to the percentage of inhibition on cell migration at 10.0 μM, except for that of 5637 cells treated by compound 43 at 5 μM, the minimum effective concentration for cell migration. ^a^ No inhibitory activities against cell migration were observed at 10.0 μM.

Compounds	Anti-Migration Activity (Im, %)		
	5637	SGC-7901	PC-3
**Andrographolide, 43**	34.5%	20.8%	N ^a^
**44**	44.7%	N ^a^	N ^a^

**Table 17 biomolecules-10-00138-t017:** SAR study of imidazobenzothiazole derivatives. ^a^ Log of molar concentration that inhibits 50% net cell growth, MG MID—mean graph midpoint.

Compounds	X	R_1_	R_2_	R_3_	R_4_	log MG MID GI_50_ (M) ^a^
**45**	S	H	H	Cl	H	−4.74
**46**	S	H	H	OCH_3_	H	−4.87
**47**	S	H	OCH_3_	OCH_3_	H	−4.14
**48**	S	H	H	Cl	Cl	−4.89
**49**	S	H	OCH_3_	OCH_3_	OCH_3_	−4.26

**Table 18 biomolecules-10-00138-t018:** SAR study of purine derivatives.

Compounds	Anti-proliferation activity (IC_50_, μM)			
	MCF-7	HCT-116	A-375	G-361
**50**	3.93 ± 0.04	6.20 ± 0.05	1.18 ± 0.03	3.06 ± 0.01
**51**	5.63 ± 0.03	6.36 ± 0.06	4.98 ± 0.07	5.67 ± 0.01

**Table 19 biomolecules-10-00138-t019:** SAR study of 8-hydroxyquinoline derivatives. ^a^ Log of molar concentration of compound that inhibits cell growth by 50%, MG MID—mean graph midpoint.

Compounds	R	log MG MID GI_50_ (M) ^a^
**52a**	4-I	1.8
**52b**	4-F	1.6
**52c**	4-Cl	0.7
**52d**	4-Br	1.3
**52e**	2,4,6-(Cl)_3_	0.7

**Table 20 biomolecules-10-00138-t020:** SAR study of (Z)-1-(1,3-diphenyl-1H-pyrazol-4-yl)-3-(phenylamino)prop-2-en-1-one derivatives.

Compounds	R_1_	R_2_	Anti-proliferation activity (IC_50_, μM)				
			HT29	PC3	A549	U87MG	HaCaT
**53**	H	Cl	3.61 ± 0.56	5.2 ± 0.93	14.05 ± 0.76	11.4 ± 0.29	28.4 ± 4.1
**54**	H	CF_3_	>50	>50	>50	>50	>50
**55**	H	OH	5.0 ± 0.96	3.6 ± 0.65	3.21 ± 1.2	4.29 ± 0.89	44.6 ± 3.6
**56**	H	OMe	1.56 ± 0.32	6.4 ± 1.1	3.25 ± 0.19	>50	>50
**57**	H	3,4,5-OMe	9.8 ± 1.3	5.93 ± 1.7	6.34 ± 0.83	2.35 ± 0.65	36.2 ± 1.8
**58**	F	Cl	4.6 ± 0.61	3.8 ± 0.56	6.7 ± 0.85	8.9 ± 1.3	21.3 ± 2.1
**59**	F	CF_3_	>50	>50	>50	>50	>50
**60**	F	OH	1.9 ± 0.21	2.6 ± 0.19	1.5 ± 0.45	4.7 ± 0.8	16.3 ± 1.2
**61**	F	OMe	3.2 ± 0.9	4.6 ± 0.7	8.9 ± 0.51	2.5 ± 0.61	19.6 ± 0.93
**62**	F	3,4,5-OMe	7.77 ± 0.96	4.89 ± 1.35	9.35 ± 1.8	12.6 ± 2.1	32.8 ± 3.4
**63**	Cl	Cl	4.65 ± 0.63	3.89 ± 0.79	3.67 ± 0.3	13.12 ± 1.2	23.4 ± 3.7
**64**	Cl	CF_3_	>50	>50	>50	>50	>50
**65**	Cl	OH	2.5 ± 0.27	4.43 ± 1.3	1.91 ± 0.21	1.50 ± 0.43	22.6 ± 2.3
**66**	Cl	OMe	8.93 ± 1.4	10.65 ± 1.1	6.46 ± 2.7	6.89 ± 1.95	30.8 ± 2.9
**67**	Cl	3,4,5-OMe	12.7 ± 2.6	9.98 ± 0.69	5.64 ± 0.56	17.8 ± 3.6	46.6 ± 5.2
**68**	OMe	Cl	4.76 ± 0.57	3.89 ± 0.33	2.97 ± 0.26	8.86 ± 0.3	20.9 ± 1.5
**69**	OMe	CF_3_	9.87 ± 0.31	>50	>50	>50	>50
**70**	OMe	OH	2.46 ± 0.57	1.98 ± 0.16	2.77 ± 0.24	3.73 ± 0.66	34.6 ± 2.5
**71**	OMe	OMe	8.76 ± 0.98	13.4 ± 1.7	6.78 ± 3.4	>50	>50
**72**	OMe	3,4,5-OMe	14.6 ± 1.7	18.9 ± 2.3	11.2 ± 1.65	>50	>50
**73**	3,4,5-OMe	Cl	7.68 ± 0.92	11.2 ± 1.43	8.67 ± 0.75	3.21 ± 0.36	30.2 ± 2.8
**74**	3,4,5-OMe	CF_3_	>50	>50	>50	>50	>50
**75**	3,4,5-OMe	OH	2.5 ± 0.09	4.6 ± 0.78	3.16 ± 0.92	1.8 ± 0.57	17.6 ± 1.1
**76**	3,4,5-OMe	OMe	5.78 ± 1.9	9.6 ± 1.7	4.78 ± 0.41	12.8 ± 2.3	>50
**77**	3,4,5-OMe	3,4,5-OMe	8.4 ± 2.63	14.1 ± 1.94	7.98 ± 1.78	5.1 ± 0.93	20.9 ± 1.5

**Table 21 biomolecules-10-00138-t021:** SAR study of naphthoquinone amide derivatives.

Compounds	R	R_1_	R_2_	Anti-Proliferation Activity (IC_50_, μM)	
				HeLa	SAS
**78a**	CH_2_Ph	CH_3_	OCH_3_	>100	56.5
**78b**	CH_2_·CH(CH_3_)_2_	CH_3_	OCH_3_	>100	78.5
**79a**	H	OCH_3_	H	77.5	12.0
**79b**	CH_3_	OCH_3_	H	39.0	14.0
**79c**	CH(CH_3_)_2_	CH_3_	H	20.0	16.0

**Table 22 biomolecules-10-00138-t022:** SAR study of 1’S-1’-acetoxychavicol acetate (ACA) derivatives.

Compounds	R_1_	R_2_	Anti-Proliferation Activity (IC_50_, μM)
ACA	-	-	4.8 ± 0.4
AEA	-	OCH_3_	9.5 ± 0.3
AMCA	OCH_3_	OCH_3_	29.6 ± 5.6

**Table 23 biomolecules-10-00138-t023:** SAR study of 6,7-disubstituted-4-phenoxyquinoline derivatives.

Compounds	R_1_	R_2_	Anti-proliferation activity (IC_50_, μMol/L)
HT-29	H460	A549	MKN-45	U87MG
**80**	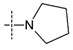	2-CF_3_	0.15 ± 0.01	0.18 ± 0.01	0.13 ± 0.01	0.09 ± 0.02	1.12 ± 0.02
**81**	4-F	0.16 ± 0.02	0.20 ± 0.03	0.14 ± 0.04	0.33 ± 0.01	1.90 ± 0.21
**82**	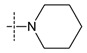	2-CF_3_	0.15 ± 0.02	0.19 ± 0.02	0.33 ± 0.03	0.08 ± 0.003	1.23 ± 0.01
**83**	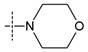	2-F	0.17 ± 0.04	0.19 ± 0.03	0.12 ± 0.02	0.09 ± 0.01	1.30 ± 0.02

**Table 24 biomolecules-10-00138-t024:** SAR study of chalcone thiosemicarbazide derivatives.

Compounds	R_1_	R_2_	R_3_	R_4_	R_5_	Anti-Proliferation Activity (IC_50_, μM)
						HepG2
**84a**	OMe	H	H	H	H	20 ± 3
**84b**	H	H	OMe	H	H	5.53 ± 0.3
**84c**	H	OMe	H	H	H	10 ± 2
**84d**	H	H	H	H	Br	6.35 ± 0.34
**84e**	H	H	H	H	Me	0.78 ± 0.05

**Table 25 biomolecules-10-00138-t025:** SAR study of (‒)-arctigenin derivatives.

Compounds	R_1_	R_2_	R_3_	Preferential Cytotoxicity (PC_50_, μM)
**Arctigenin, 85**	Me	Me	Me	0.80
**86a**	Me	Et	Et	0.66
**86b**	Et	Me	Me	0.49
**86c**	Et	Et	Et	0.78

**Table 26 biomolecules-10-00138-t026:** SAR study of asiatic acid derivatives.

Compounds	R	Anti-Proliferation Activity (IC_50_, μM)
HT-29	HeLa
**Asiatic acid, 87**	-	64.30 ± 3.21	52.47 ± 0.06
**88**	H	51.25 ± 1.77	60.17 ± 2.75
**89**	-	23.50 ± 0.71	24.50 ± 1.41
**90**	-	1.28 ± 0.08	1.08 ± 0.04
**91**	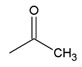	2.02 ± 0.19	1.40 ± 0.14
**92**	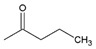	1.29 ± 0.09	0.95 ± 0.01
**93**	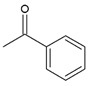	1.05 ± 0.05	0.80 ± 0.04
**94**	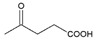	6.35 ± 0.64	3.48 ± 0.04
**95**	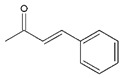	0.71 ± 0.02	0.67 ± 0.07
**96**	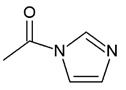	2.37 ± 0.23	1.62 ± 0.09
**97**	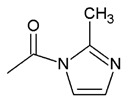	2.80 ± 0.14	1.63 ± 0.15
**98**	-	N.D.	7.50 ± 0.42
Cisplatin	-	N.D.	2.28 ± 0.26

**Table 27 biomolecules-10-00138-t027:** Summary of the antimigration and antiproliferation effects of different substituents.

**Effective Functional Groups for Antimigration Effect**			
**Functional Groups**	**Analogues of**	**Reference**	**Figure**
Fluoro	Isocoumarin	[9]	Figure 1
	Brartemicin	[10]	Figure 2
	1,3,4-oxadiazole	[12]	Figure 3
	Benzoylbenzophenone thiosemicarbazone	[15]	Figure 4
Methoxy	4-anilino-quinazoline	[23]	Figure 5
	2-furanylvinylquinoline	[24]	Figure 6
	4-methyl-2-(4-pyridinyl)thiazole-5-carboxamide	[27]	Figure 7
	EF24	[28]	Figure 8
Methyl	BU-4664L	[29]	Figure 9
	Isomalyngamide A	[31]	Figure 10
	Diarylamino-1,3,5-triazine	[32]	Figure 11
Amino	4-anilino-quinazoline	[23]	Figure 5
	Triarylethylene	[33]	Figure 12
	Benzamide Ilomastat	[38]	Figure 13
Hydroxy	Brartemicin	[10]	Figure 2
Nitro	4-methyl-2-(4-pyridinyl)thiazole-5-carboxamide	[27]	Figure 7
Bromo	2,3-diaryl-4-thiazolidinone	[39]	Figure 14
Chloro	Benzyloxyphenylmethylaminophenol	[44]	Figure 15
**Weak Functional Groups for Antimigration Effect**			
**Functional Groups**	**Analogues of**	**Reference**	**Figure**
Methoxy	Brartemicin	[10]	Figure 2
	Benzoylbenzophenone thiosemicarbazone	[15]	Figure 4
	Curcumin	[45]	Figure 16
Methyl	Brartemicin	[10]	Figure 2
	1,3,4-oxadiazole	[12]	Figure 3
	Sipholenol A	[46]	Figure 17
Hydroxy	Benzoylbenzophenone thiosemicarbazone	[15]	Figure 4
	Andrographolide	[47]	Figure 18
Bromo	Benzoylbenzophenone thiosemicarbazone	[15]	Figure 4
Chloro	4-methyl-2-(4-pyridinyl)thiazole-5-carboxamide	[27]	Figure 7
Methylamino	Diarylamino-1,3,5-triazine	[32]	Figure 11
**Effective Functional Groups for Antiproliferation Effect**			
**Functional Groups**	**Analogues of**	**Reference**	**Figure**
Chloro	Imidazobenzothiazole	[48]	Figure 19
	Purine	[49]	Figure 20
	8-hydroxyquinoline	[50]	Figure 21
Methoxy	(Z)-1-(1,3-diphenyl-1H-pyrazol-4-yl)-3-(phenylamino)prop-2-en-1-one	[51]	Figure 22
	Naphthoquinone amide	[52]	Figure 23
Fluoro	1,3,4-oxadiazole	[12]	Figure 3
	6,7-disubstituted-4-phenoxyquinoline	[54]	Figure 24
Methyl	Naphthoquinone amide	[52]	Figure 2
	Thiosemicarbazide	[55]	Figure 25
Hydroxy	(Z)-1-(1,3-diphenyl-1H-pyrazol-4-yl)-3-(phenylamino)prop-2-en-1-one	[51]	Figure 22
Ethoxy	(‒)-arctigenin	[56]	Figure 26
Carbonyl	Fluorinated asiatic acid	[57]	Figure 27
**Weak Functional Groups for Antiproliferation Effect**			
**Functional Groups**	**Analogues of**	**Reference**	**Figure**
Methoxy	Imidazobenzothiazole	[48]	Figure 19
	Thiosemicarbazide	[55]	Figure 25
Bromo	8-hydroxyquinoline	[50]	Figure 21
	Thiosemicarbazide	[55]	Figure 25
Methyl	1,3,4-oxadiazole	[12]	Figure 3
Fluoro	8-hydroxyquinoline	[50]	Figure 21
Iodo	8-hydroxyquinoline	[50]	Figure 21
Chloro	(Z)-1-(1,3-diphenyl-1H-pyrazol-4-yl)-3-(phenylamino)prop-2-en-1-one	[51]	Figure 22
Trifluoromethyl	(Z)-1-(1,3-diphenyl-1H-pyrazol-4-yl)-3-(phenylamino)prop-2-en-1-one	[51]	Figure 22
Hydroxy	Fluorinated asiatic acid	[57]	Figure 27

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
