# Peer review of "A Review of the Structure–Activity Relationship of Natural and Synthetic Antimetastatic Compounds"

_biomolecules, 2020, doi:10.3390/biom10010138_

Round 1

Reviewer 1 Report

The manuscript biomolecules-595147 is devoted to the actual problem of medicinal chemistry. The reviewed article is interesting and theme of the article meets the scope of the journal. Work is performed at sufficient scientific level and has good quality; the results of literature review are professionally interpreted. However, it needs minor revision before publication.

To improve the quality and perception of the manuscript I would suggest paying attention to following comments:

In Figures 1 (page 2) and 6 (page 6), tables should be removed because they will only represent one compound. Moreover, information in table is duplicated in the main text. In figure 14 (page 12), the structural formula of the thiazolidinone derivative must be written in full, because in the presented form it can be interpreted as acid bromide. References list should be carefully checked and journal style policy should be strictly followed (Abbreviated Journal Name, DOI etc). There are grammar/typing and orthographical errors in the manuscript.

After correction this manuscript can be accepted for publication. My decision is minor revision.

Reviewer 2 Report

Reviewed work has big potential, but is prepared in untidy manner and need some corrections. In particular:

Be consequent in numbering of presented compounds. In some places described compounds and their derivatives or analogues are numbered 12, 13, 14 ... etc, in another place 24a, 24b, 24c.  Fig. 3. Why IC50 for anti-proliferative activity is μg/ml, whereas for FAK inhibitory activity μM. Fig. 4. Cathepsin, not capthsin nor capthesin. Line 109 - compound 22c, Fig. 6 - compound 22. Is that correct? Line 120 - probably twice, not one fold Fig. 10 - Anti-proliferative activity (IC50), in table values in percent [%] - it is inconsequential Fig. 16 Bisdemethoxycurcumin Fig. 18 Different numbers below structures and in the table Line 334, 383 etc. "and friends" is not acceptable, use "and co-workers", "et al."

Authors must read carefully whole text again and correct all these lapses.

Reviewer 3 Report

The title isn't clear,  I suggest A review on the structure-activity relationship of  natural and synthetic compounds for anti-metastatic effect.

Round 2

Reviewer 2 Report

My comments and suggestions were considered, manuscript is suitable for publication.

Author Response

Thank you